# RadD from *Fusobacterium nucleatum* engages NKp46 to promote antitumor cytotoxicity

**Ahmed Rishiq[1†], Johanna Galaski[1,2†], Reem Bsoul[2†], Mingdong Liu[1], Rema Darawshe[3], Renate Lux[4], Gilad Bachrach[2‡], Ofer Mandelboim[1*‡]**

[1]The Concern Foundation Laboratories at the Lautenberg Center for Immunology and Cancer Research, Institute for Medical Research Israel Canada (IMRIC), Hebrew University Hadassah Medical School, Jerusalem, Israel; [2]Institute of Medical Microbiology and Hygiene, Medical Centre University of Freiburg, Freiburg, Germany; [3]The Institute of Dental Sciences, The Hebrew University-Hadassah School of Dental Medicine, Jerusalem, Israel; [4]Section of Biosystems and Function, Division of Oral and Systemic Health Sciences, UCLA School of Dentistry, Los Angeles, United States

**\*For correspondence:**
oferm@ekmd.huji.ac.il

[†]These authors contributed equally to this work

[‡]These authors jointly supervised this work

## eLife Assessment

This **useful** study describes a mechanism of microbial modulation of anti-tumor immunity, which is of considerable interest in the field. However, the experimental supports for the key mechanistic claim, the interaction between RadD and NKp46, are not robust. Multiple experimental inconsistencies, especially in vivo, weaken the conclusions, making the strength of evidence **incomplete**. Additional controls, direct binding assays, and clarification of in vivo mechanistic relevance would strengthen the work.

**Abstract** *Fusobacterium nucleatum*, a gram-negative bacterium implicated in periodontal disease, contributes to tumor progression in various cancers. Whether the presence of *F. nucleatum* inhibits tumor progression of some cancers is largely unknown. Here, we identify an interaction between *F. nucleatum* and the natural killer (NK) cell receptor NKp46. Analysis of TCGA datasets revealed that the co-occurrence of *F. nucleatum* and high NKp46 expression correlates with improved survival in head and neck cancers but not in colorectal cancers. Using binding assays, we demonstrate that both human NKp46 and its murine ortholog, Ncr1, directly recognize the fusobacterial adhesin RadD. Genetic deletion of *radD* or blockade of NKp46 significantly impaired NK cell-mediated cytotoxicity in vitro and promoted tumor-cell growth. In vivo, infection with *F. nucleatum* accelerated tumor progression, with an exacerbated effect observed in the absence of RadD or NKp46. These findings highlight RadD as a critical ligand for NKp46 and establish the NKp46–RadD axis as a key interface in host–microbe–tumor interactions, offering a novel target for immunotherapeutic intervention in cancer influenced by microbial factors.

## Introduction

*Fusobacterium nucleatum*, a gram-negative bacterium, has received considerable interest in recent years due to its significance in various human diseases, including cancer (*Alon-Maimon et al., 2022*). This anaerobic bacterium is mostly found in the human oral cavity (*de Andrade et al., 2019*). Recent

studies showed that *F. nucleatum* influence the progression of various tumor types through its interactions with the host immune system, modulation of inflammatory pathways, and potential involvement in metastatic processes (*Guo et al., 2024*; *Parhi et al., 2020*). *F. nucleatum* not only facilitates but also actively promotes cancer proliferation and metastasis through established mechanisms, such as invasion of epithelial and endothelial cells via its virulence factors, as well as through pathways that remain to be elucidated (*Guo et al., 2024*; *Parhi et al., 2020*). In the context of immune interactions, it was shown that the *F. nucleatum* Fusobacterial apoptosis-inducing protein 2 (Fap2) interacts with the TIGIT receptor on natural killer (NK) cells and T cells, leading to inhibition of NK cell cytotoxicity and T cell activity (*Gur et al., 2015*). Additionally, another adhesin, CbpF, was found to bind CEACAM1 on T cells, modulating their activity (*Galaski et al., 2021*). We also demonstrated that the RadD protein of *Fusobacterium nucleatum* subsp. *nucleatum* interacts with SIGLEC7 on NK cells, leading to the suppression of NK cell-mediated killing of cancer cells (*Galaski et al., 2024*). In contrast, however, the identity of the *F. nucleatum* ligands that interact with NK activating receptors and how NK cell recognizes this bacterium is still poorly understood.

The natural killer (NK) cell receptor NKp46 plays a significant role in immune response regulation, particularly in the identification and eradication of infected or transformed cells (*Barrow et al., 2019*). NKp46 was shown to recognize and bind hemagglutinins in both the influenza and the parainfluenza viruses (*Mandelboim et al., 2001*). Heparan sulfate (HS) and some bacterial and fungal proteins were also identified as ligands for NKp46 (*Barrow et al., 2019*). Moreover, NKp46 was found to recognize an externalized calreticulin (ecto-CRT), which translocated from the ER to the cell membrane during ER stress (*Sen Santara et al., 2023*). NKp46 was also shown to interact with *F. nucleatum* in the oral cavity (*Chaushu et al., 2012*). However, the *F. nucleatum* ligand that is recognized by NKp46 and whether the interaction between NKp46 and *F. nucleatum* is important in cancer development and patient's prognosis remains currently unknown.

Here we demonstrate that the *F. nucleatum* RadD adhesin is a ligand for NKp46 and that this interaction plays a significant role in tumor development.

## Results

### NKp46 expression modifies the prognostic impact of *F. nucleatum* in a tumor-type-specific manner

To evaluate the prognostic significance of *F. nucleatum* in the context of NKp46 activity, we analyzed transcriptomic data from The Cancer Genome Atlas (TCGA) alongside microbial abundance profiles from The Cancer Microbiome Atlas (TCMA) across two tumor types. In head and neck squamous cell carcinoma (HNSC), patients exhibiting both *F. nucleatum* positivity and expression of NKp46 (encoded by NCR1) had significantly improved overall survival compared to NKp46[+] patients lacking *F. nucleatum* (log-rank p<0.05; *Figure 1A*, *Figure 1—source data 1A*). Conversely, in colorectal cancer (CRC), *F. nucleatum* status did not significantly affect survival among NKp46[+] patients (*Figure 1B*, *Figure 1—source data 1B*). The median survival in the *F. nucleatum*[+]NKp46[+] HNSC subgroup was 5.81 years, compared to 2.36 years in the *F. nucleatum*[-]NKp46[+] group, corresponding to a hazard ratio (HR) of 2.08 (95% CI: 1.20–3.61; *Figure 1C*). In CRC, median survival was similar between *F. nucleatum*[+]NKp46[+] patients (5.15 years) and their *F. nucleatum*[-] counterparts (5.85 years), with no significant difference in risk (HR = 0.71, 95% CI: 0.26–1.95; *Figure 1C*). Importantly, NKp46 expression levels were substantially higher in HNSC compared to CRC (*Figure 1*, *Figure 1—source data 1D*), suggesting a possible threshold-dependent effect of NKp46 on microbial–immune interactions. We also analyzed bulk RNA expression datasets for *SIGLEC7* and *CEACAM1* and evaluated their associations with HNSC and CRC using the same patient databases utilized in our study (*Figure 1—figure supplement 1*). No significant differences in *SIGLEC7* expression were detected between HNSC and CRC samples (*Figure 1—figure supplement 1A*, *Figure 1—source data 1*). Although *CEACAM1* mRNA levels did not differ between *F. nucleatum*-positive and -negative cases, its overall expression was higher in CRC compared to HNSC (*Figure 1—figure supplement 1B*, *Figure 1—source data 1*). Together, these findings underscore a tumor-type-specific interplay between microbial colonization and immune contexture, positioning NKp46 as a key modulator of *F. nucleatum*-associated clinical outcomes.

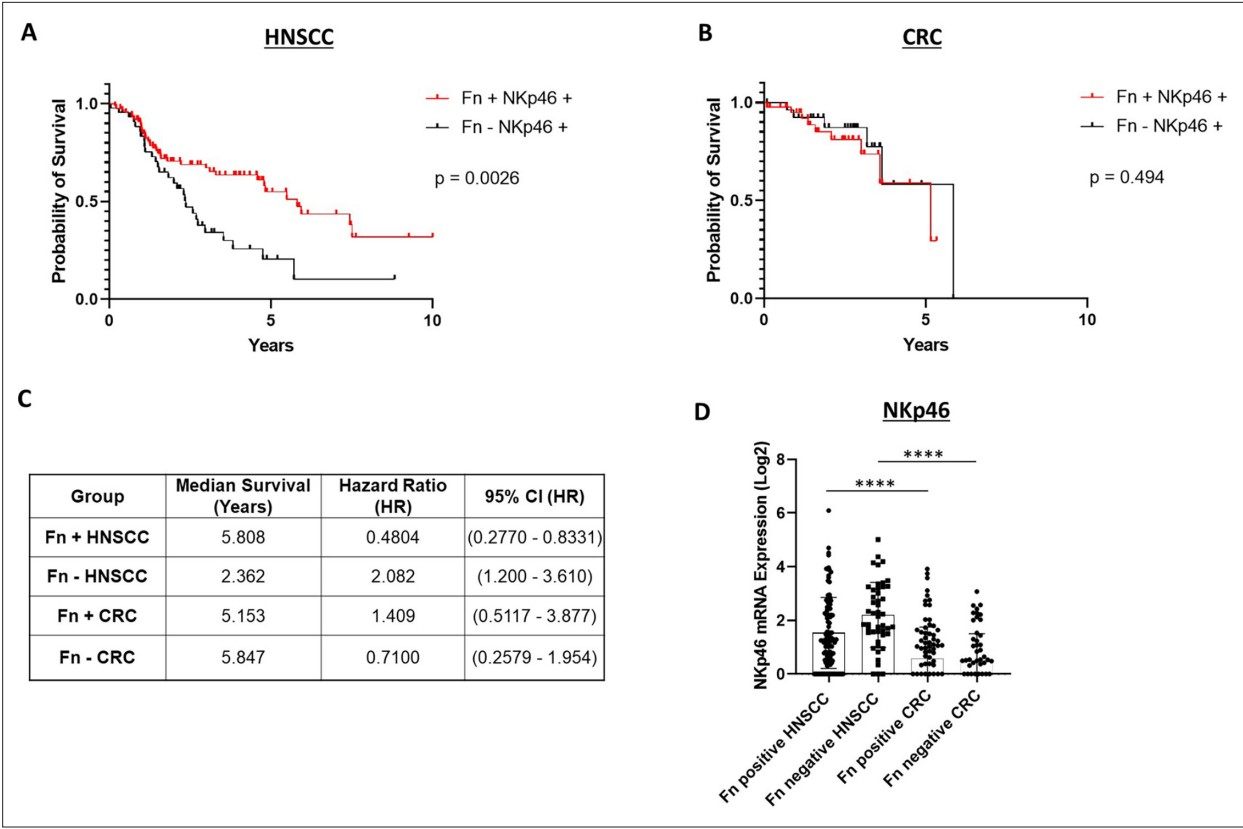

**Figure 1.** NKp46 expression modifies the prognostic effect of *F. nucleatum* in a tumor-type-specific manner. (**A**) Kaplan–Meier survival curves for head and neck squamous cell carcinoma (HNSC) patients stratified by *F. nucleatum* status and NKp46 (NCR1) expression. Patients with concurrent *F. nucleatum* positivity (n=87) and high NKp46 expression exhibited significantly improved overall survival compared to those who were *F. nucleatum*-negative (n=44) but NKp46-positive (log-rank p<0.05). (**B**) Kaplan–Meier survival curves for colorectal cancer (CRC) patients stratified by the same criteria showed no significant difference in survival between *F. nucleatum*-positive (n=44) and *F. nucleatum*-negative (n=31) groups. (**C**) Table summarizing hazard ratios (HR) for *F. nucleatum*-negative cases among NKp46[+] patients. In HNSC, absence of *F. nucleatum* was associated with a significantly poorer prognosis (HR = 2.08, 95% CI: 1.20–3.61), whereas in CRC, *F. nucleatum* absence showed no significant association with patient prognosis (HR = 0.71, 95% CI: 0.26–1.95). (**D**) Comparison of NKp46 expression across HNSC and CRC tumors. Log$_2$ expression levels of NKp46 mRNA were compared across HNSC and CRC cohorts, stratified by *F. nucleatum* positive and negative. Results were analyzed by one-way ANOVA with Bonferroni post hoc correction. ****p≤0.0001.

The online version of this article includes the following source data and figure supplement(s) for figure 1:

**Source data 1.** *Figure 1A* HNSCC and *Figure 1B* CRC survival raw data.

**Figure supplement 1.** *SIGLEC7* and *CEACAM1* expression and the prognostic effect of *F. nucleatum* in a tumor-type-specific manner.

**Figure supplement 1—source data 1.** The raw data for the SIGLEC7 and CEACAM1 mRNA expression and the prognostic effect of *F. nucleatum* in a tumor-type-specific manner and statistical tests used and significance.

## NKp46 binds RadD

We previously reported that NKp46 interacts with *F. nucleatum* and that this interaction plays a role in the context of periodontal disease (*Chaushu et al., 2012*). However, the identity of the *F. nucleatum* ligand recognized by NKp46 has remained unknown. Given that the co-occurrence of NKp46 expression and *F. nucleatum* presence correlates with improved prognosis in certain tumors (*Figure 1*), identifying the specific NKp46-binding ligand on *F. nucleatum* is of critical importance for understanding the underlying mechanisms and potential therapeutic implications of this interaction. To identify the *F. nucleatum* ligand of NKp46, we assessed the binding of NKp46 Ig, its D1 domain (D1 Ig), its mouse orthologue Ncr-1 Ig, and CD16 Ig to FITC-labeled *F. nucleatum* strains ATCC 10953 and ATCC 23726, which represent the subspecies *polymorphum* and *nucleatum*, respectively (*Figure 2*). Surprisingly, NKp46, its D1 domain, and the mouse Ncr-1 exhibited higher binding to ATCC 10953 compared to ATCC 23726 (*Figure 2A*), while little or no binding was observed for the CD16, which was used as a

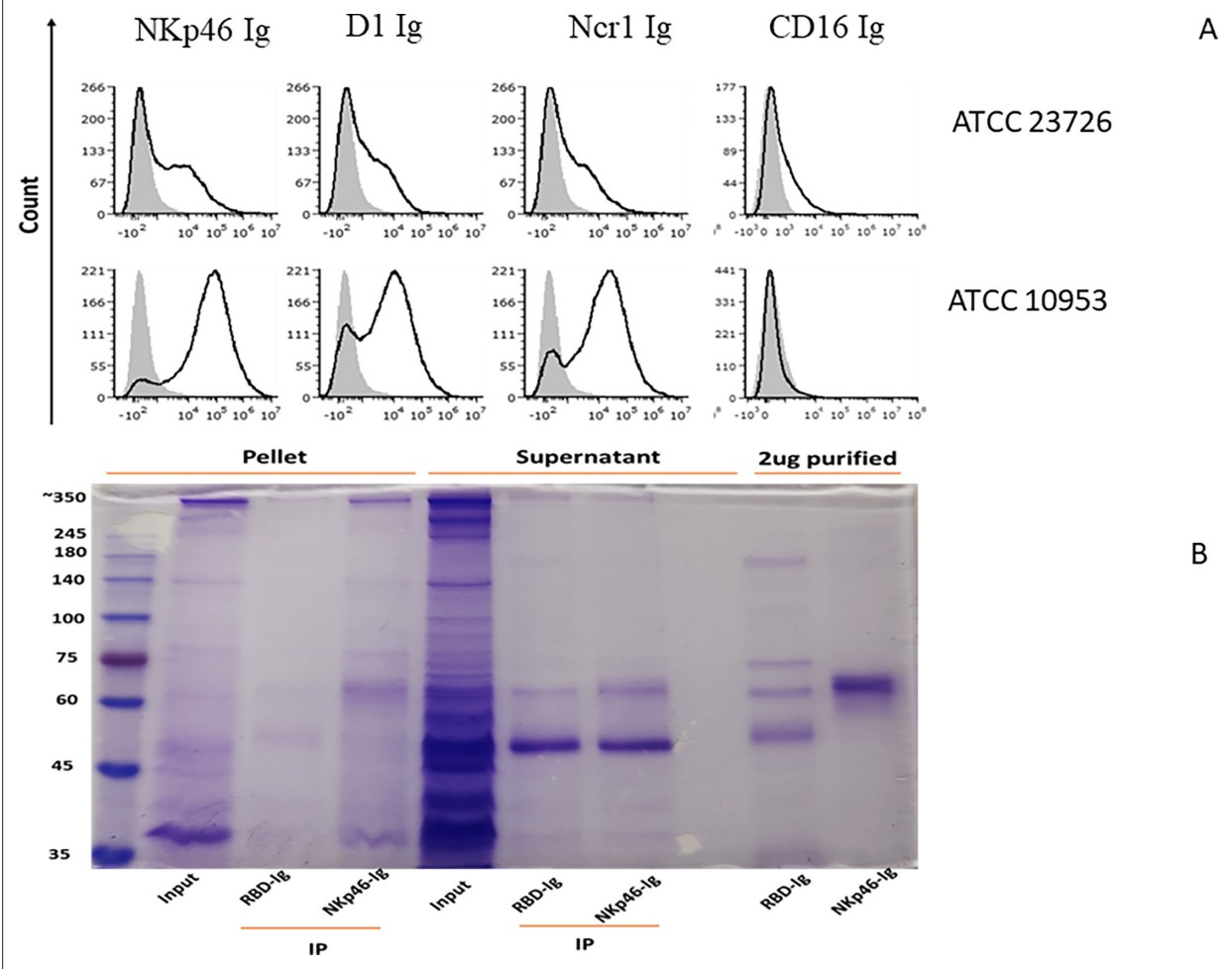

**Figure 2.** Binding of *Fusobacterium nucleatum* to NKp46 and its D1 domain. (**A**) The figure shows histograms of FITC-labeled *F. nucleatum* subsp. *nucleatum* ATCC 23726 (upper histograms) and ATCC 10953 (lower histograms) incubated with 2 µg of NKp46 Ig, D1 domain of NKp46 (D1 Ig), Ncr-1 Ig, and CD16 Ig fusion proteins. Representative staining from one of two independent experiments is shown. (**B**) An immunoprecipitation assay was performed using an NKp46–Ig fusion protein with *F. nucleatum* subsp. *nucleatum* ATCC 23726. Molecular weight markers are shown on the left. Lane 1 (Input) contains total lysates from the bacterial pellet (membrane protein fraction), showing a band corresponding to RadD (~350 kDa)-arrow. Lane 2 (RBD–Ig control) shows immunoprecipitation with 2.5 µg of control RBD–Ig, with no detectable band at ~350 kDa. Lane 3 (NKp46–Ig) shows immunoprecipitation with 2.5 µg of NKp46–Ig, revealing a band at ~350 kDa-arrow. Lanes 4–6 correspond to the supernatant fraction of the bacterial lysate. No bands are observed in lanes 5 and 6, indicating a lack of interaction in this fraction. Lanes 7 and 8 contain 2.5 µg of purified RBD–Ig and NKp46–Ig proteins, respectively.

The online version of this article includes the following source data and figure supplement(s) for figure 2:

**Source data 1.** NKp46 immunoprecipitation with *Fusobacterium nucleatum* ATCC 23726 lysates.

**Source data 2.** An original Western blot gel as well as the labeled and uncropped one, which was used in validating the interaction between Fusobacterium nucleatum and NKp46.

**Figure supplement 1.** Ccm-1 Ig binding to *Fusobacterium ATCC 23726* and *ATCC 10953*.

control. To further control the experiments, we used another fusion protein, mouse Ceacam1 (Ccm-1 Ig) that, unlike its human homologue, does not interact with CbpF (*Galaski et al., 2021*). Across multiple independent experiments, Ccm-1 Ig binding did not differ significantly between the two bacterial strains, and comparable binding levels were observed (*Figure 2—figure supplement 1*). NKp46 Ig and Ncr-1 Ig, however, displayed similar binding profiles (*Figure 2—figure supplement 1*).

To validate the interaction between *Fusobacterium* and NKp46, we performed immunoprecipitation assays using an NKp46 Ig fusion protein with *F. nucleatum* subsp. *polymorphum* ATCC 10953 and *F. nucleatum* subsp. *nucleatum* ATCC 23726. Immunoprecipitations were unsuccessful (not shown) when using *F. nucleatum* subsp. *polymorphum* ATCC 10953 (reasons are unknown). However, when

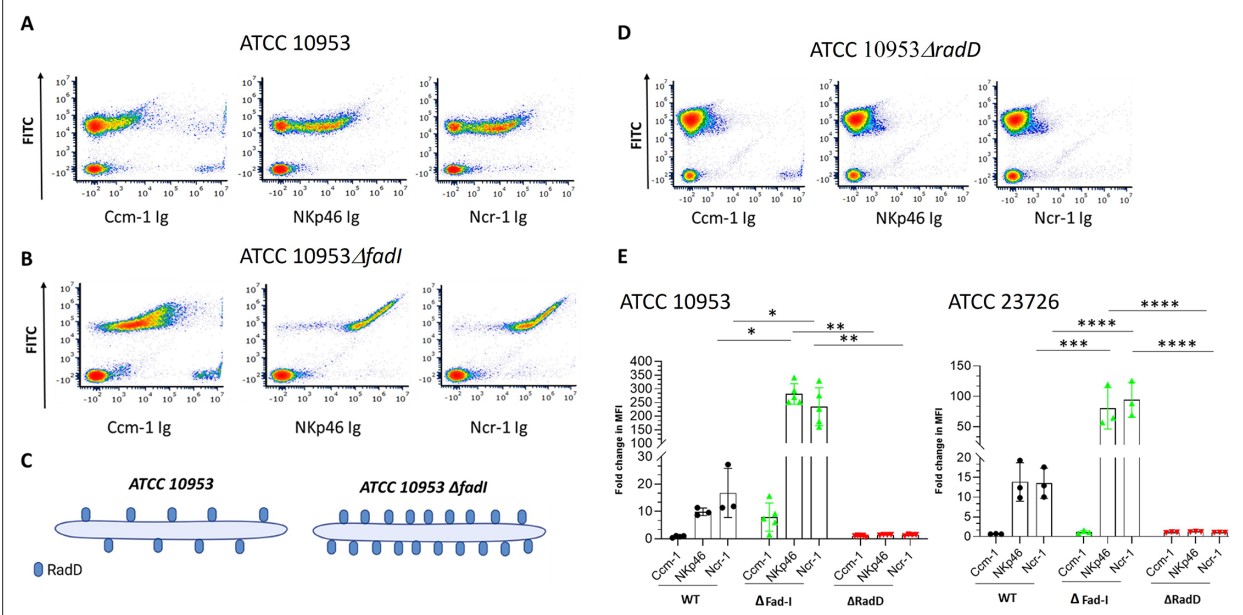

**Figure 3.** RadD is the bacterial ligand for NKp46. (**A, B**) Density plot of FITC labeled ATCC 10953 (**A**) and its *ΔfadI* mutant derivative ATCC 10953 *ΔFad-I* (**B**) stained with the various fusion proteins (listed in the X axis). (**C**) Schematic representation of ATCC 10953 wild type (WT) strain and RadD surface expression (left) compared to ATCC 10953 *ΔFad-I* (right). (**D**) Density plot of the FITC-labeled *ΔRadD* mutant strain of ATCC 10953 stained with various fusion proteins (listed in the X axis). The figure shows data from one representative experiment out of three to five independent experiments. (**E**) Fold change quantification of FITC-labeled bacteria binding to the fusion proteins Ccm1-Ig, NKp46 Ig, and Ncr-1 Ig in ATCC 10953 (left) and ATCC 23726 (right). Summary of three to five independent experiments. The mean value ± SD of the experiments is presented. *p<0.05, **p≤0.01, ***p≤0.001, and ****p≤0.0001.

The online version of this article includes the following source data for figure 3:

**Source data 1.** Raw data for median fluorescent intensity (MFI) measurements corresponding to *Figure 3E*, including quantification for *Fusobacterium nucleatum* strains ATCC 10953 and ATCC 23726, as described in the *Figure 3E* legend.

using the *F. nucleatum* subsp. *nucleatum* ATCC 23726 strain, NKp46 Ig precipitated a protein of approximately 350 kDa (*Figure 2B*, *Figure 2—source data 1*), suggesting that the *Fusobacterium* ligand recognized by NKp46 has a mass of around 350 kDa.

While we were staining various FITC-labeled *F. nucleatum* deletion-mutated strains with NKp46 to try and identify the NKp46 ligand, we observed an unexpected elevation in binding of NKp46 Ig and Ncr-1 Ig to the FadI deleted mutant of *F. nucleatum* subsp. *polymorphum* ATCC 10953 (ATCC 10593 *ΔFadI*), while the expression of a control Ig fusion protein Ccm-1 Ig was only minimally increased (please compare *Figure 3A and B*). Since we showed previously that the absence of FadI results in overexpression of RadD (illustrated in *Figure 3C* and *Shokeen et al., 2020*), we incubated Ccm-1 Ig, NKp46 Ig, and Ncr-1 Ig with a FITC-labeled ATCC 10953 mutant lacking the major multifunctional adhesin RadD (ATCC 10593 *ΔRadD*). Interestingly, we observed that the lack of RadD abolished fuso-bacterial binding of NKp46 Ig and Ncr-1 Ig (*Figure 3D*).

Quantification of binding levels confirmed that the absence of FadI resulted in a 250–300-fold increase in binding of NKp46 Ig and Ncr-1 Ig to ATCC 10953 *ΔFad-I,* while the absence of RadD almost completely abolished NKp46 Ig and Ncr-1 Ig interaction (*Figure 3E*, left, *Figure 3—source data 1*). A similar effect was observed for the corresponding mutant strains of ATCC 23726, albeit at a lower level, with ATCC 23726 *Δfad-I* exhibiting an about 100-fold increase, which is abolished in the absence of RadD (*Figure 3E*, right, *Figure 3—source data 1*). These findings indicate that the autotransporter protein RadD is the ligand of NKp46.

## Arginine and anti-NKp46 antibody inhibit the binding of NKp46 to RadD

Since RadD is an arginine-inhibitable adhesin (*Kaplan et al., 2009*), we tested whether arginine can block the binding of RadD to NKp46 or its mouse ortholog Ncr-1. Incubation of FITC-labeled ATCC

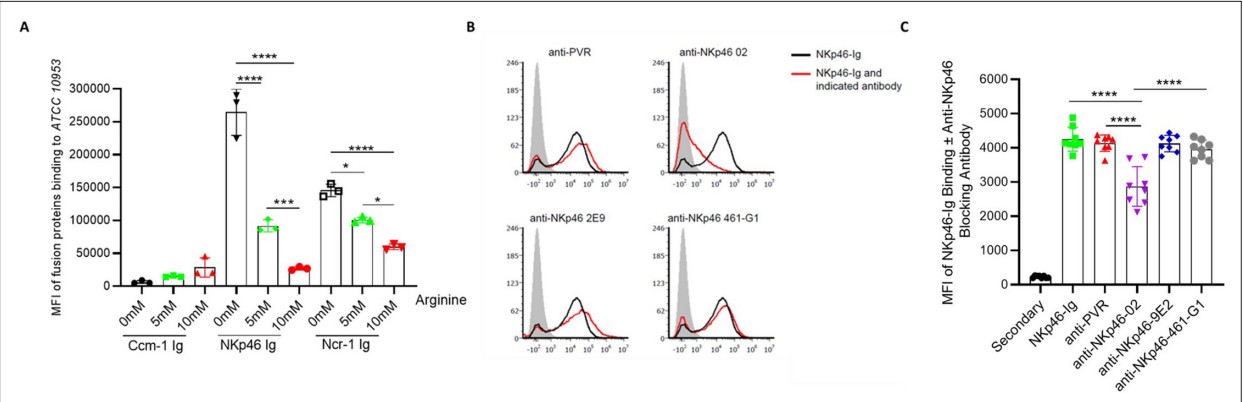

**Figure 4.** NKp46-02 antibody and arginine block ATCC 10953 binding to NKp46. (**A**) Quantification of median fluorescent intensity (MFI) of FITC-labeled ATCC 10953 binding to Ccm1-Ig, NKp46-Ig, and Ncr-1 Ig, without or with 5 and 10 mM of L-Arginine. Data combined from three to four independent experiments are presented. (**B**) NKp46-Ig (2 μg) was pre-incubated with 1 μg of a control anti-PVR antibody and NKp46 monoclonal antibodies (9E2, 461-G1, and 02) to evaluate the blocking of ATCC 10953 interaction with the NKp46 receptor. (**C**) Shows the quantification results of histograms depicted in (**B**). The mean value ± SD of the experiments (n=8) is presented. *p<0.05, **p≤0.01, ***p≤0.001, and ****p≤0.0001.

The online version of this article includes the following source data and figure supplement(s) for figure 4:

**Source data 1.** Median fluorescence intensity (MFI) quantification of fusion protein binding to *Fusobacterium nucleatum* ATCC 10953 corresponding to *Figure 4A*.

**Figure supplement 1.** Fusobacterium binding inhibition by L-Arginine.

**Figure supplement 2.** Arginine inhibition of NKp46-Ig and Ncr1-Ig binding in *F. nucleatum* ΔFadI.

**Figure supplement 2—source data 1.** Raw data showing the mean fluorescence intensity (MFI) for ATCC 10953 ΔFadI (A) and ATCC 23726 ΔFadI (B), including the corresponding statistical analyses and significance levels.

10953 with arginine followed by staining with NKp46 Ig and Ncr-1 Ig revealed that arginine inhibits NKp46 and Ncr-1 Ig binding in a dose-dependent manner (*Figure 4—source data 1*), whereas Ccm-1 Ig binding was minimally affected (*Figure 4*, represented histograms are depicted in *Figure 4—figure supplement 1*). Similar observations were noted for the *F. nucleatum* ΔFadI mutants of *F. polymorphum* ATCC 10953 and *F. nucleatum* ATCC 23726 (*Figure 4—figure supplement 2—source data 1*).

NKp46 consists of two extracellular domains, a membrane-distal (D1) domain and a membrane-proximal (D2) domain (*Barrow et al., 2019*). Previous studies indicate that the vast majority of NKp46 ligands are recognized through the D2 domain (*Arnon et al., 2004*). Interestingly, *F. nucleatum* seemed to be recognized by the D1 domain of the NKp46 receptor (*Figure 2*). To further examine whether the D1 domain of NKp46 is involved in its binding to *F. nucleatum,* we used several anti-human NKp46 antibodies (461-G1, hNKp46.02 (02), and 9E2) that were previously shown to bind the D1 domain of NKp46 (*Berhani et al., 2019*). We pre-incubated NKp46 Ig individually with all of these antibodies prior to addition to ATCC 10953. Notably, no difference in binding of ATCC 10953 was observed when the NKp46 Ig was incubated with either 9E2, 461-G1, or anti-PVR antibody, which served as controls (*Figure 4B*). However, blocking with hNKp46.02 (02) antibody significantly reduced the NKp46 Ig-ATCC 10953 interaction (*Figure 4B*, quantified in *Figure 4—source data 1C*). These findings confirm that the NKp46 receptor interacts with ATCC 10953 specifically via its D1 domain.

## NKp46-RadD interactions lead to tumor cell killing in vitro and in vivo

Next, we examined the impact of the NKp46.02 (02)-blocking antibody on NK cell cytotoxicity against tumor cells incubated with *F. nucleatum*. We co-incubated or not NK cells with the 02 antibody. Subsequently, NK cells were co-cultured with human mammary gland carcinoma cell lines MCF7 and T47D that were pre-incubated with ATCC 10953 or with the corresponding ΔRadD mutant strain. NK cytotoxicity was assessed using the Calcein-AM assay (illustrated in *Figure 5A*). We observed that the 02 antibody had no effect on NK cell cytotoxicity against the breast cancer cell lines in the absence of *F. nucleatum* (*Figure 5*). However, in the presence of *F. nucleatum* ATCC 10953, a significant increase in tumor cell killing (approximately 1.2–1.5-fold change) was noticed for unblocked NK cells compared to 02 blocked NK cells (*Figure 5B and C*). Interestingly, this increased NK cytotoxicity was diminished

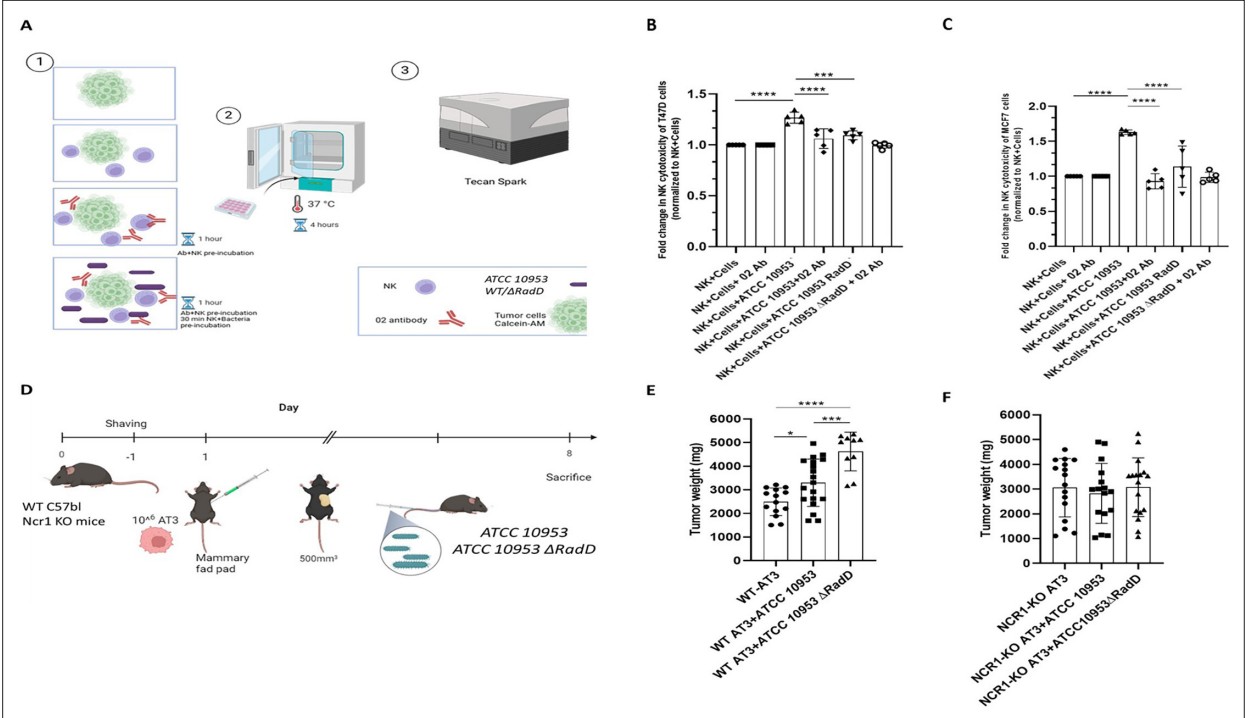

**Figure 5.** Cytotoxicity and tumor growth is RadD and Ncr1-dependent. (**A**) Schematic diagram showing the design of the NK cells cytotoxicity assay against breast cancer cell lines T47D and MCF7. 1. Tumor cells were stained with Calcein-AM dye and then incubated either with tumor cells (T47D or MCF7) only, tumor + NK, tumor +bacteria (ATCC 10953 WT and ATCC 10953 ΔRadD)+NK with/without preincubation with 02 antibody. 2. Killing assays were performed in a 37°C incubator for 4 hours. 3. The fluorescence intensity of Calcein was measured to determine cell viability using a spectrophotometer (Tecan Spark). Summary of NK cytotoxicity against T47D (**B**) and MCF7 (**C**) breast cancer cell lines. Combined results from five independent experiments. (**D**) C57BL/6 or NCR1-KO mice were shaved and AT3 cells (1 × 10⁶ cells in 100 μl PBS) were injected 1 day later into the mammary fat pad. When tumors reached a size of about 500 mm³, mice were inoculated intravenously with 5 × 10⁷ ATCC 10953 WT and 5 × 10⁷ ATCC 10953 ΔRadD bacteria. Eight days later, mice were sacrificed and tumor weight was determined. (**E**) The tumor weight of C57BL/6 or NCR1-KO (**F**) mice. The figure shows the combination of 4–5 experiments performed. The mean value ± SD of the experiments is presented. NK + cells + ATCC10953 RadD is the ATCC10953 deleted for RadD. *p<0.05, **p≤0.01, ***p≤0.001, and ****p≤0.0001.

The online version of this article includes the following source data for figure 5:

**Source data 1.** Raw data for cytotoxicity quantification shown in *Figure 5B and C*.

---

when the tumor cells were incubated with ATCC 10953 ΔRadD, as in the absence of RadD, NKp46-blocking had no effect (*Figure 5B* and *Figure 5—source data 1C*).

To test whether the NKp46-RadD interactions are important for controlling tumor growth, we established a syngeneic mouse breast cancer model by implanting the AT3 cell line orthotopically in the mammary fat pad of C57BL/6 wild type (WT) and Ncr-1 deficient mice (NCR-1 KO). The tumor was allowed to grow to reach approximately 500 mm³ in volume prior to intravenous inoculation with either ATCC 10953 or ATCC 10953 ΔRadD. Mice were sacrificed on day 8 following the bacterial injection, and tumor weight was measured (illustrated in *Figure 5D*). The tumor weight was significantly increased in tumor-bearing WT mice inoculated with ATCC 10953 as compared to uninfected tumor-bearing WT mice (*Figure 5E*). Strikingly, a further increase in tumor weight was observed when mice were injected with ATCC 10953 ΔRadD mutated bacterium (*Figure 5—source data 1E*). The increased tumor growth was not observed in the NCR-1 KO mice (*Figure 5—source data 1F*). These results suggest that *RadD* recognition by the NKp46 activating receptor is required for better cytotoxicity against tumors infected with *F. nucleatum*.

## Discussion

Patient prognosis is a key determinant in the association between *F. nucleatum* infection and tumor development and progression in colorectal cancer (*Lee et al., 2021*). Our analysis of TCGA PanCancer

and TCMA datasets revealed that the co-occurrence of *F. nucleatum* and NKp46 expression in head and neck squamous cell carcinoma (HNSC) is associated with a protective effect. In contrast, no such association was observed in colorectal cancer cohorts, likely due to the reduced NKp46 expression levels in these patients (*Cerami et al., 2012*; *de Bruijn et al., 2023*; *Dohlman et al., 2021*; *Gao et al., 2013*).

In this study, we demonstrate that the RadD outer-surface protein of *F.nucleatum* is specifically recognized by the D1 domain of the NKp46 receptor. This interaction enhances the ability of NK cells to kill tumor cells infected with *F. nucleatum*. Previous studies have shown that *F. nucleatum* promotes tumor growth in colorectal cancer (*Zhu et al., 2024*) and in breast cancer in mice (*Parhi et al., 2020*), and suppresses immune cell activity through engagement of three inhibitory receptors: CEACAM1, TIGIT, and Siglec-7 (*Galaski et al., 2024*; *Galaski et al., 2021*; *Gur et al., 2015*). Furthermore, it was also shown that CD147, which is overexpressed on the surface of colorectal cancer (CRC) cells, also binds to RadD. The binding of RadD to CD147 leads to the enrichment of *F. nucleatum* within CRC tissues, triggering an oncogenic cascade PI3K–AKT–NF-κB signaling pathway, which promotes tumor-igenesis (*Zhang et al., 2024*). Here, we uncover a counterbalancing mechanism, whereby NK cells counter this immune suppression through NKp46-mediated recognition of RadD.

*F. nucleatum* RadD is an outer membrane autotransporter protein that mediates binding between *F. nucleatum* and other oral bacteria by promoting interspecies interactions, which facilitates dental plaque development and virulence in periodontal disease (*Kaplan et al., 2009*). RadD, in combina-tion with Fap2, has been identified as virulence factors capable of inducing cell death in lymphocytes (*Kaplan et al., 2010*). As an adhesin, RadD also facilitates the coaggregation of *F. nucleatum* with *Clostridioides difficile* (*C. difficile*). This interaction promotes biofilm formation within the intestinal mucus, potentially contributing to the pathogenesis of *C. difficile* infection (*Engevik et al., 2021*).

*Fusobacterium*-associated defensin inducer (Fad-I) is a cell wall-associated diacylated lipoprotein of *F. nucleatum* that acts as a key microbial molecule enhancing the host's innate immune response at mucosal surfaces by promoting human beta-defensin (hBD-2) expression (*Bhattacharyya et al., 2016*). Inactivation of the *fad-I* (Δ*fad-I*) gene results in a significant increase in *radD* gene expression, leading to elevated *radD* transcript levels, and subsequently, the binding to *Streptococcus gordonii* is increased (*Shokeen et al., 2020*).

While NKp46 showed strong binding to *F. nucleatum* subsp. *polymorphum* (ATCC 10953), less binding was observed for *F. nucleatum* subsp. *nucleatum* (ATCC 23726). The reasons for this are still unknown due to the lack of RadD specific antibodies; however, one likely explanation might be differ-ences in RadD expression between the different *F. nucleatum* subspecies.

Because deletion of *F. nucleatum fad-I* resulted in enhanced binding of both human and mouse NKp46 receptors, and since *fad-I* deletion resulted in elevated surface expression of RadD (*Shokeen et al., 2020*), we hypothesized that RadD is a ligand for NKp46. Indeed, NKp46 Ig precipitated a protein band at the size of RadD, and the NKp46/Ncr1 binding was markedly reduced to the *F. nucleatum* Δ*RadD* strain. Consistent with previous studies (*Kaplan et al., 2009*), we observed that NKp46 binding to RadD is also arginine-inhibitable; however, the arginine effect on Ncr1 binding was less pronounced. Ncr-1, the murine orthologue of human NKp46, shares approximately 58% sequence identity with its human counterpart (*Biassoni et al., 1999*). Thus, these arginine-dependent differences might stem from structural differences or distinct posttranslational modifications, such as glycosylation. Indeed, prediction algorithms combined with high-performance liquid chromatography analysis revealed that Ncr-1 possesses two putative novel O-glycosylation sites, of which only one is conserved in humans (*Glasner et al., 2015*).

Using the NKp46 blocking antibody (02), which targets the D1 domain of the receptor (*Berhani et al., 2019*), we demonstrated that this antibody effectively disrupts *F. nucleatum* binding to NKp46 and impairs NK cell-mediated killing against tumor cells. In our in vivo model, WT mice infected with the ATCC 10953Δ*RadD* strain exhibited significantly greater tumor weight relative to those infected with the wild-type ATCC 10953 strain. Intriguingly, infections with either the wild-type ATCC 10953 or ATCC 10953 Δ*RadD* strains were not able to affect tumor progression in NCR-1 KO mice. These results collectively support our hypothesis that NK cell cytotoxicity is mediated by the presence of NKp46 on NK cells and the expression of RadD on the *F. nucleatum* surface (*Figure 6*).

Our findings align with those of another group that investigated the Δ*RadD* mutant in a mouse model of preterm birth (*Wu et al., 2021*). Using the ATCC 23726 strain, the authors revealed an earlier

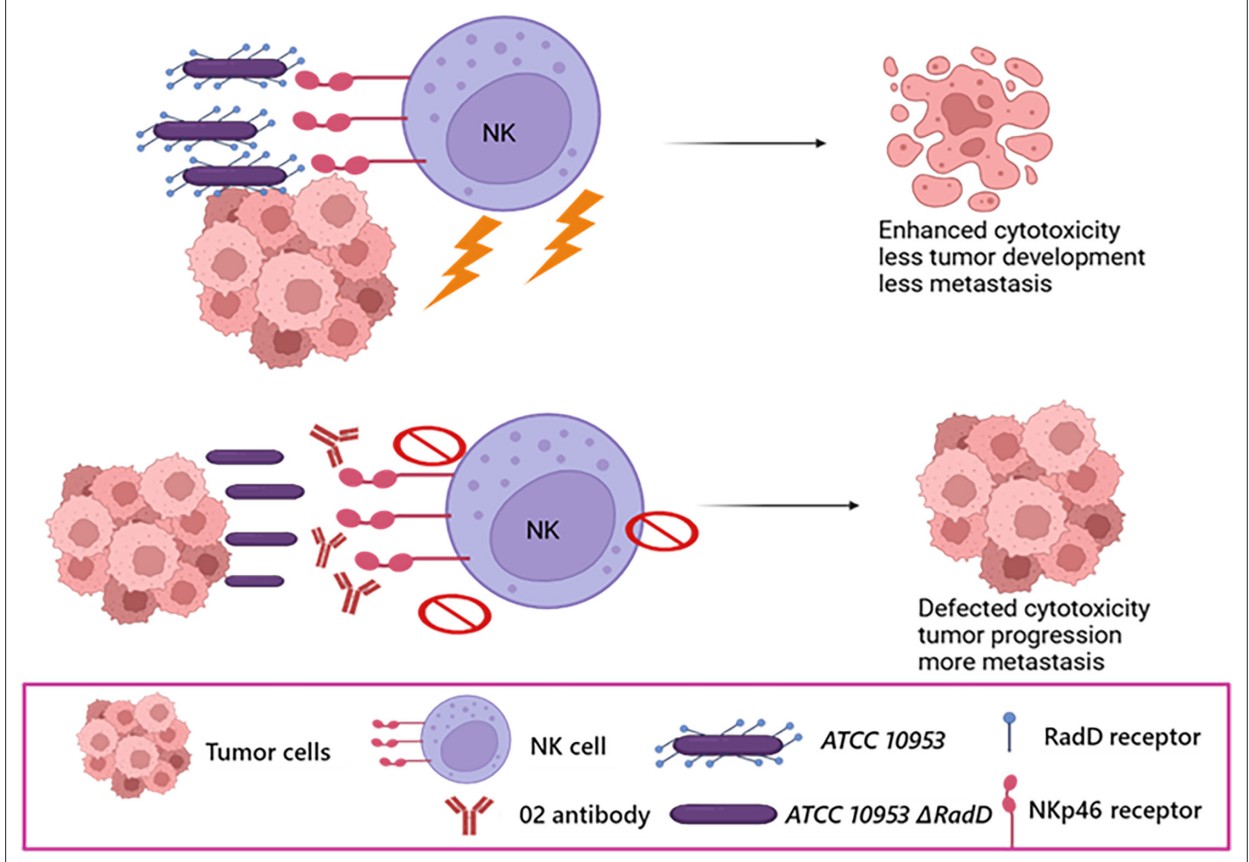

**Figure 6.** Postulated model for the RadD-NKp46 interaction impact on NK cytotoxicity and tumor growth. NKp46 interaction with RadD expressed by *Fusobacterium nucleatum* triggers NK cell cytotoxicity. This activation enhances tumor cell killing in vitro and in vivo. Conversely, the absence of RadD or the blocking of NKp46 impairs NK cell activity, leading to tumor. This figure was created using BioRender.com.

and increased invasion of the *ΔRadD* strain, which reached the placenta, amniotic fluid, and fetus sooner, and continued accumulating over time in comparison with the WT *F. nucleatum*. Moreover, the RadD mutant exhibited reduced systemic clearance, with no decline observed in liver or spleen levels, suggesting impaired immune evasion or a lack of control over dissemination (*Wu et al., 2021*).

In conclusion, we demonstrate that *F. nucleatum* RadD functions as a direct ligand for NKp46 and highlight its critical role in modulating NK cell activity both in vitro and in vivo. Strategies aimed at elevating NKp46 expression or enhancing its activity might further strengthen NK cell responses and help reduce cancer development associated with *F. nucleatum* infection.

## Materials and methods

**Key resources table**

| Reagent type (species) or resource | Designation | Source or reference | Identifiers | Additional information |
|---|---|---|---|---|
| Antibody | APC α-human NKp46 'mouse monoclonal' | Biolegend | Cat#331917; RRID:AB_2561649 | 0.2 ug |
| Antibody | Purified anti-human CD335 (NKp46) Antibody (Clone 9E2) 'mouse monoclonal' | Biolegend | Cat#331902; RRID:AB_1027637 | 0.2 ug |
| Antibody | α-human NKp46- 461-G1 'mouse monoclonal' | In-house *Berhani et al., 2019* | In-house | 2 ug for blocking |

*Continued on next page*

*Continued*

| Reagent type (species) or resource | Designation | Source or reference | Identifiers | Additional information |
|---|---|---|---|---|
| Antibody | α-human NKp46- 02mAb 'mouse monoclonal' | In-house *Berhani et al., 2019* | In-house | 2 ug for blocking |
| Antibody | Human CD155/PVR Antibody 'mouse monoclonal' | R&D Systems | Catalog #: MAB25301 RRID:AB_2174021 | 2 ug for blocking |
| Antibody | Alexa Fluor 647 AffiniPure F(ab') Fragment Donkey Anti-Human IgG | Jackson ImmunoResearch | Cat#709-606-098; RRID:AB_2340580 | (1:200) |
| Antibody | Alexa Fluor 647 AffiniPure F(ab') Fragment Goat Anti-Mouse IgG (H+L) | Jackson ImmunoResearch | Cat#115-606-146; RRID:AB_2338930 | (1:200) |
| Antibody | PE Mouse IgG1, κ Isotype Ctrl Antibody (MOPC-21) 'mouse monoclonal' | Biolegend | Cat#400112; RRID:AB_2847829 | 0.2 ug |
| Antibody | APC Mouse IgG1, κ Isotype Ctrl Antibody (MOPC-21) 'mouse monoclonal' | Biolegend | Cat#400120; RRID:AB_2888687 | 0.2 ug" |
| Other | *F. nucleatum* ATCC23726 | ATCC | N/A | *F. nucleatum* strain maintained in Ofer Mandelboim's lab |
| Other | *F. polymorphum* ATCC 10953 | ATCC | N/A | *F. nucleatum* strain maintained in Ofer Mandelboim's lab |
| Other | *F. nucleatum* ATCC 23726 ΔRadD | *Kaplan et al., 2009*, Mol Microbiol | N/A | *F. nucleatum* strain maintained in Ofer Mandelboim's lab |
| Other | *F. polymorphum* ATCC 10953 ΔRadD | *Guo et al., 2024* Mol Oral Microbiol | N/A | *F. nucleatum* strain maintained in Ofer Mandelboim's lab |
| Other | *F. nucleatum* ATCC 23726 Δfadl | *Shokeen et al., 2020* Microorganisms | N/A | *F. nucleatum* strain maintained in Ofer Mandelboim's lab |
| Other | *F. polymorphum* ATCC 10953 Δfadl | *Bhattacharyya et al., 2016* | N/A | *F. nucleatum* strain maintained in Ofer Mandelboim's lab |
| Peptide, recombinant protein | NKp46 fusion protein-(NKp46 Ig) (human) | In-house | N/A | |
| Peptide, recombinant protein | Ncr-1 fusion protein (Ncr-1 Ig) (mouse) | In-house | N/A | |
| Peptide, recombinant protein | D1 domain of NKp46 fusion protein- (D1 Ig) | In-house | N/A | |
| Peptide, recombinant protein | CD16 fusion protein- CD16 Ig | In-house | N/A | |
| Peptide, recombinant protein | Ccm-1 fusion protein- Ccm-1 Ig | In-house | N/A | |
| Cell line (*Homo-sapiens*) | HEK293T | In-house | ATCC: CRL-3216 | Human embryonic kidney cell line Cell line maintained in Ofer Mandelboim's lab |
| Cell line (*Homo sapiens*) | MCF7 | In-house | ATCC: HTB-22 | Breast cancer cell line Cell line maintained in Ofer Mandelboim's lab |
| Cell line (*Homo sapiens*) | T47D | In-house | ATCC: CRL-2865 | Breast cancer cell line Cell line maintained in Ofer Mandelboim's lab |
| Commercial assay or kit | Fluorescein-Isothiocyanate Isomer I (FITC) | Sigma | Cat#7250 | |
| Commercial assay or kit | Arginine | Sigma | Cat#A8094 | |

*Continued on next page*

*Continued*

| Reagent type (species) or resource | Designation | Source or reference | Identifiers | Additional information |
|---|---|---|---|---|
| Commercial assay or kit | Calcein AM | Thermo Fisher | Cat#C1413 | |
| Commercial assay or kit | Protein A/G-Sepharose affinity Chromatography | (Sigma) | GE17-0405-01 | |
| Commercial assay or kit | EasySep Human NK Cell Isolation Kit | (STEMCELL Technologies) | Cat#17955 | |
| Software, algorithm | Prism 8 | GraphPad | RRID:SCR_002798 | |
| Software, algorithm | FCS Express | De Novo Software | RRID:SCR_016431 | |
| Software, algorithm | BioRender | | RRID:SCR_018361 | |
| Other (female mice) | C57BL/6 | Envigo | RRID:MGI:2159769 | C57BL/6 inbred mice (C57BL/6JOlaHsd) https://www.inotiv.com/research-model/c57bl-6jolahsd |
| Other (female mice) | Ncr-1 | In-house *Gazit et al., 2006* Nature Immunology | RRID:MGI:5699740 | See the results section in *Gazit et al., 2006* |

## TCGA and TCMA

RNA-sequencing expression data and corresponding clinical and survival information were retrieved from the PanCancer Atlas dataset available through cBioPortal (https://www.cbioportal.org/). Data for head and neck squamous cell carcinoma (HNSC) and colorectal cancer (CRC) were selected for analysis. Microbial abundance scores for *F. nucleatum* were curated from The Cancer Microbiome Atlas (TCMA). Co-occurrence of NKp46 expression and *F. nucleatum* abundance was evaluated using survival analyses Kaplan–Meier with log-rank test using Prism were Statistical significance was defined as $p<0.05$.

## Cell lines

C57BL/6 mouse mammary carcinoma cell line (AT3), human breast mammary gland adenocarcinoma MCF7, and T47D cell lines were cultured with DMEM or RPMI with 10% inactivated fetal bovine serum (Sigma-Aldrich), 1 mM sodium pyruvate (Biological Industries), 2 mM glutamine (Biological Industries), nonessential amino acids (Biological Industries), 100 U/ml penicillin (Biological Industries), and 0.1 mg/ml streptomycin (Biological Industries). All cells were tested regularly for mycoplasma using Mycolor One-Step Mycoplasma Detector (Vazyme) kit. Primary NK cells were isolated from the human peripheral blood of healthy individuals using EasySep Human NK Cell Isolation Kit (STEMCELL Technologies) and then cultured in F12-DMEM medium supplemented with 10% human serum (Sigma-Aldrich), 1 mM sodium pyruvate (Biological Industries), 2 mM glutamine (Biological Industries), nonessential amino acids (Biological Industries), 100 U/ml penicillin (Biological Industries), 0.1 mg/ml streptomycin (Biological Industries), and 400 IU of recombinant human hIL2 (Peprotech).

## Fusion proteins

To generate fusion proteins, the extracellular portion of the protein of interest was cloned into a mammalian expression vector containing the mutated Fc portion of human IgG1 (CSI-Ig IRES-Puro Fc mut N197A). Fusion proteins NKp46 Ig, CD16-Ig, D1-Ig, Ncr-1 Ig were generated in HEK293T cells and purified using Protein A/G-Sepharose affinity Chromatography (Sigma-Aldrich).

## Bacteria cultivation

The bacterial strains used in this study were *F. nucleatum* subsp. *nucleatum* ATCC 23726, *F. nucleatum* subsp. *polymorphum* ATCC 10953, and their respective ΔRadD and ΔfadI mutant derivatives (*Lee et al., 2021*). Bacteria were kept in −80°C frozen glycerol stocks and grown at 37°C on blood agar plates (Hylabs) under anaerobic conditions generated using the Oxoid AnaeroGen anaerobic gas

generator system (Thermo Fisher). Bacteria were harvested from blood agar plates for subsequent experimental procedures.

## Bacteria staining and flow cytometry

In brief, for bacteria staining experiments, bacteria were harvested from blood agar plates, washed twice with PBS (Sartorius), and incubated with 0.1 mg/ml FITC (Sigma-Aldrich) in PBS at room temperature in the dark for 30 minutes on a shaker. Subsequently, bacteria were washed thrice in PBS at 4000 rpm for 10 minutes to remove unbound FITC. Next, bacteria were divided into 96-well U plates at 2 million bacteria per well and incubated with 2 μg of fusion proteins per well for 1 hour on ice, followed by washing and 30 minutes of incubation with Alexa Fluor 647-conjugated donkey anti-human IgG (Jackson ImmunoResearch). Histograms of bacteria were gated on FITC-positive cells. The mean fluorescence intensity (MFI) fold change was calculated by dividing the MFI obtained from staining with the fusion proteins by the MFI of the corresponding secondary antibody control (bacteria incubated without fusion proteins).

For arginine-blocking experiments, incubation of *ATCC 10953* was performed in the presence of arginine (5 mM or 10 mM) for 30 minutes. Subsequently, 2 ug of Ccm-1 Ig, NKp46 Ig, or Ncr-1Ig were added for another 30 minutes. Bacteria were centrifuged at 4000 rpm for 10 minutes, washed with PBS, and stained with Alexa Fluor 647-conjugated donkey anti-human IgG (Jackson ImmunoResearch) for 30 minutes on ice.

Antibody-blocking experiments were performed by incubating 2 μg of NKp46 Ig with 1 μg of α-human NKp46 (clones 9E2, 461-G1, and 02) or anti-PVR antibodies for 1 hour on ice. Subsequently, this incubation was followed by washing 2 times with PBS and 30 minutes with Alexa Fluor 647-conjugated donkey anti-human IgG.

## Immunoprecipitation

Fusobacteria were lysed using RIPA buffer, and the lysates were centrifuged twice to separate the supernatant from the pellet (which contains the bacterial membranes). The resulting lysates were incubated overnight with 2.5 μg of purified NKp46 and protein G-beads. After thorough washing, the bound proteins were placed in the sample buffer and heated at 95°C for 8 minutes. The eluates were run on a 10% acrylamide gel and visualized by Coomassie blue staining.

## In vitro cytotoxicity

NK cytotoxicity was investigated as previously described (*Galaski et al., 2024*). We found that for *ATCC 10953 ΔRadD* an MOI of 25:1 was necessary to achieve adhesion to the breast cancer cell line that was similar to MOI of 10:1 for the control *ATCC 10953* strain. Effector NK cells (100,000 cells) isolated from healthy individuals were incubated with or without 2 ug of anti-NKp46 (02) antibody in antibiotic-free RPMI medium for 1 hour on ice. Then, *ATCC 10953* and *ATCC 10953 ΔRadD* were incubated with NK cells for 30 minutes in a 37°C incubator. Calcein-AM (Thermo Scientific) stained tumor cell lines were co-cultured with the effector NK cells in a 10:1 effector-to-target ratio for an additional 4 hours in a 37°C incubator. The maximal killing was determined by adding Triton-X (9.5 ml RPMI + 0.5 ml Triton-X) to the target cells (with or without bacteria), and the spontaneous release was determined by adding only target cells (with or without bacteria). Plates were centrifuged (1600 rpm for 5 min, 4°C), and supernatants (75 μl) were transferred to a black 96-well plate. The Calcein-AM release into the supernatant was measured using a Tecan Spark multiplate reader with excitation/emission wavelengths at 485 nm/535 nm. Specific lysis percentage was calculated as follows: $\frac{(\text{Experimental} - \text{Spontaneous lysis})}{(\text{Maximal} - \text{Spontaneous lysis})} \times 100$. The fold change was calculated by normalizing the experimental groups to the NK and tumor cells (with or without 02 antibody).

## In vivo experiments

All in vivo experimental procedures were approved by the Hebrew University of Jerusalem committee with the Ethical Approval Number of Research MD-21-16479-5 and conducted in the specific pathogen-free rooms (SPF) of the animal facility according to the guidelines of the Institutional Animal Care & Use Committee (IACUC).

7–8-week-old female wild-type C57BL/6 and Ncr-1 knockout mice (NCR-1 KO) were injected orthotopically (mammary fat pad) with $1 \times 10^6$ AT3 tumor cells. At a tumor size of 500 mm³, mice

were randomly divided into three groups and injected intravenously with $5 \times 10^7$ *F. nucleatum* ATCC 10953, $7.5 \times 10^7$ *F. nucleatum* ATCC 10953 *ΔRadD*, and one group of AT3 tumor cells only. Mice were sacrificed at day 8, and tumor weight was measured.

## Statistical analysis

Statistical analysis and graphs were prepared using GraphPad Prism version 8 (GraphPad Software). Comparison among groups was performed by one-way ANOVA multiple comparison test followed by Tukey's post hoc test. p-value was considered significant at $p < 0.05$.

## Acknowledgements

This work was supported by the following grants awarded to OM: the ICRF grant, the ISF grant, the Israeli Innovation Authority grants 72670 and 75934, and the ISF grant 307/22 and ICRF Project Grant to GB.

## Additional information

### Funding

| Funder | Grant reference number | Author |
|---|---|---|
| Israel Science Foundation | 307/22 | Gilad Bachrach |
| Israel Cancer Research Fund | | Gilad Bachrach Ofer Mandelboim |
| Israel Science Foundation | 619/23 | Ofer Mandelboim |
| Israeli Innovation Authority | 75934 | Gilad Bachrach Ofer Mandelboim |
| Israel Science Foundation (IPMP) | 3042/22 | Ofer Mandelboim Gilad Bachrach |
| Israeli Innovation Authority | 72670 | Gilad Bachrach Ofer Mandelboim |

The funders had no role in study design, data collection and interpretation, or the decision to submit the work for publication.

### Author contributions

Ahmed Rishiq, Conceptualization, Resources, Data curation, Validation, Investigation, Visualization, Methodology, Writing – original draft, Writing – review and editing; Johanna Galaski, Conceptualization, Data curation, Validation, Investigation, Visualization, Methodology, Writing – original draft, Writing – review and editing; Reem Bsoul, Rema Darawshe, Data curation, Investigation, Methodology; Mingdong Liu, Data curation, Validation, Investigation, Methodology; Renate Lux, Data curation, Supervision, Investigation, Writing – original draft, Writing – review and editing; Gilad Bachrach, Conceptualization, Resources, Data curation, Supervision, Funding acquisition, Validation, Investigation, Methodology, Writing – original draft, Writing – review and editing; Ofer Mandelboim, Conceptualization, Data curation, Formal analysis, Supervision, Funding acquisition, Validation, Investigation, Methodology, Writing – original draft, Writing – review and editing

### Author ORCIDs

Ahmed Rishiq ⓘ https://orcid.org/0000-0001-6564-1050
Reem Bsoul ⓘ https://orcid.org/0009-0004-0124-1954
Gilad Bachrach ⓘ https://orcid.org/0000-0003-1350-2280
Ofer Mandelboim ⓘ https://orcid.org/0000-0002-9354-1855

### Ethics

All animal experiments were conducted in accordance with the guidelines of the Hebrew University-Hadassah Medical School Institutional Animal Care and Use Committee (IACUC) and were approved under protocol number MD-21-16479-5. All procedures complied with Israeli national regulations for

animal experimentation and with the ARRIVE guidelines. Mice were housed under specific pathogen-free conditions, and all efforts were made to minimize suffering, including the use of humane endpoints and monitoring by trained veterinary staff.

Reviewer #1 (Public review): https://doi.org/10.7554/eLife.108439.2.sa1
Reviewer #2 (Public review): https://doi.org/10.7554/eLife.108439.2.sa2
Author response https://doi.org/10.7554/eLife.108439.2.sa3

---

## Additional files

### Supplementary files
MDAR checklist

### Data availability
All data generated or analysed during this study are included in the manuscript and supporting files; source data have been provided for all figures, and any additional information is available from the corresponding authors upon reasonable request.

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
