## [Editor Report · eLife Assessment]

This **useful** study describes a mechanism of microbial modulation of anti-tumor immunity, which is of considerable interest in the field. However, the experimental supports for the key mechanistic claim, the interaction between RadD and NKp46, are not robust. Multiple experimental inconsistencies, especially in vivo, weaken the conclusions, making the strength of evidence **incomplete**. Additional controls, direct binding assays, and clarification of in vivo mechanistic relevance would strengthen the work.

---

## [Referee Report · Reviewer #1 (Public review)]

In this manuscript, Rishiq et al. investigate whether natural killer (NK) cells can interact with *Fusobacterium nucleatum* and identify the molecular mediators involved in this interaction. The authors propose that the bacterial adhesin RadD may bind to the activating NK cell receptor NKp46 (NCR1 in mice), leading to NK cell activation and tumor control. While the topic is of significant interest and the hypothesis intriguing, the manuscript lacks critical experimental evidence, contains several technical concerns, and requires substantial revisions.

Major Concerns:

(1) Lack of Direct Evidence for RadD-NKp46 Interaction

The central claim that RadD interacts with NKp46 is not formally demonstrated. A direct binding assay (e.g., Biacore, ELISA, or pull-down with purified proteins) is essential to support this assertion. The absence of this fundamental experiment weakens the mechanistic conclusions of the study.

(2) Figure 2: Binding Specificity and Bacterial Strains

A CEACAM1-Ig control should be included in all binding experiments to distinguish between specific and non-specific Ig interactions. There is differential Ig binding between strains ATCC 23726 and 10953. The authors should quantify RadD expression in each strain to determine if the difference in binding is due to variation in RadD levels.

(3) Figure 3: Flow Cytometry Inconsistencies and Missing Controls

What do the FITC-negative, Ig-negative events represent? The authors should clarify whether these are background signals, bacterial aggregates, or debris.

Panel B, CEACAM1-Ig binding appears markedly increased compared to WT bacteria. The reason for this enhancement should be discussed-does it reflect upregulation of the bacterial ligand or an artifact of overexpression? Fluorescence compensation should be carefully reviewed for the NKp46/NCR1-Ig binding assays to ensure that the signals are not due to spectral overlap or nonspecific binding. Importantly, binding experiments using the FadI/RadD double knockout strain are missing and should be included. This control is essential.

In Panel E, the basis for calculating fold-change in MFI is unclear. Please indicate the reference condition to which the change is normalized.

(4) Figure 4: Binding Inhibition and Receptor Sensitivity

Panel A lacks representative FACS plots and is currently difficult to interpret. Differences in the sensitivity of human vs. mouse NKp46 to arginine inhibition should be discussed, given species differences in receptor-ligand interactions. What are the inhibition results using *F. nucleatum* strains deficient in FadI?

In Panel B, CEACAM1-Ig and RadD-deficient bacteria must be included as negative controls for binding specificity upon anti-NKp46 blocking.

(5) Figure 5: Functional NK Activation and Tumor Killing

In Panels B and C, the key control condition (NK cells + anti-NKp46, without bacteria) is missing. This is needed to evaluate if NKp46 recognition is involved in tumor killing. The authors should explicitly test whether pre-incubation of NK cells with bacteria enhances their anti-tumor activity. Alternatively, could bacteria induce stress signals in tumor cells that sensitize them to NK killing? This distinction is critical.

(6) Figure 5D: Mechanism of Peripheral Activation

It is suggested that contact between bacteria and NK cells in the periphery leads to their activation. Can the authors confirm whether this pre-activation leads to enhanced killing of tumor targets, or if bacteria-tumor co-localization is required? The literature indicates that *F. nucleatum* localizes intracellularly within tumor cells. If so, how is RadD accessible to NKp46 on infiltrating NK cells?

(8) Figure 5E and In Vivo Relevance

Surprisingly, *F. nucleatum* infection is associated with increased tumor burden. Does this reflect an immunosuppressive effect? Are NK cells inhibited or exhausted in infected mice (TGIT, SIGLEC7...)? If NK cell activation leads to reduced tumor control in the infected context, the role of RadD-induced activation needs further explanation. RadD-deficient bacteria, which do not activate NK cells, result in even poorer tumor control. This paradox needs to be addressed: how can NK activation impair tumor control while its absence also reduces tumor control?

(9) NKp46-Deficient Mice: Inconsistencies

In Ncr1⁻/⁻ mice, infection with WT or RadD-deficient *F. nucleatum* has no impact on tumor burden. This suggests that NKp46 is dispensable in this context and casts doubt on the physiological relevance of the proposed mechanism. This contradiction should be discussed more thoroughly.

---

## [Referee Report · Reviewer #2 (Public review)]

Summary:

In the present study, Rishiq et al. investigated whether the RadD protein expressed by *Fusobacterium nucleatum* subsp. Nucleatum serves as a natural ligand for the NK-activating receptor NKp46, and whether RadD-NKp46 interaction enhances NK cell cytotoxicity against tumor cells. To address this, the authors first performed an association analysis of *F. nucleatum* abundance and NKp46 expression in head and neck squamous cell carcinoma (HNSC) and colorectal cancer (CRC) using the TCMA and TCGA databases, respectively. While a positive association between NKp46⁺ and *F. nucleatum*⁺ status with improved overall survival was observed in HNSC patients, no such correlation was found in CRC.

Next, they examined the binding of NKp46-Ig to various *F. nucleatum* strains. To confirm that this interaction was mediated specifically by RadD, they employed a RadD-deficient mutant strain. Finally, to establish the functional relevance of the RadD-NKp46 interaction in promoting NK cell cytotoxicity and anti-tumor responses, they utilized a syngeneic mouse breast cancer model. In this setup, AT3 cells were orthotopically implanted into the mammary fat pad of C57BL/6 wild-type (WT) or Ncr1-deficient (NCR1⁻/⁻; murine orthologue of human NKp46) mice, followed by intravenous inoculation with either WT F. nucleatum or the ∆RadD mutant strain.

Strengths:

A notable strength of the work is that it identifies a previously unrecognized activating interaction between *F. nucleatum* RadD and the NK cell receptor NKp46, demonstrating that the same bacterial protein can engage distinct NK cell receptors (activating or inhibitory) to exert context-dependent effects on anti-tumor immunity. This dual-receptor insight adds depth to our understanding of *F. nucleatum*-immune interactions and highlights the complexity of microbial modulation of the tumor microenvironment.

Weaknesses:

(1) A previous study by this group (PMID: 38952680) demonstrated that RadD of *F. nucleatum* binds to NK cells via Siglec-7, thereby diminishing their cytotoxic potential. They further proposed that the RadD-Siglec-7 interaction could act as an immune evasion mechanism exploited by tumor cells. In contrast, the present study reports that RadD of *F. nucleatum* can also bind to the activating receptor NKp46 on NK cells, thereby enhancing their cytotoxic function.

While *F. nucleatum*-mediated tumor progression has been documented in breast and colon cancers, the current study proposes an NK-activating role for *F. nucleatum* in HNSC. However, it remains unclear whether tumor-infiltrating NK cells in HNSC exhibit differential expression of NKp46 compared to Siglec-7. Furthermore, heterogeneity within the NK cell compartment, particularly in the relative abundance of NKp46⁺ versus Siglec-7⁺ subsets, may differ substantially among breast, colon, and HNSC tumors. Such differences could have been readily investigated using publicly available single-cell datasets. A deeper understanding of this subset heterogeneity in NK cells would better explain why *F. nucleatum* is passively associated with a favorable prognosis in HNSC but correlates with poor outcomes in breast and colon cancers.

(2) The in vivo tumor data (Figure 5D-F) appear to contradict the authors' claims. Specifically, Figure 5E suggests that WT mice engrafted with AT3 breast tumors and inoculated with WT *F. nucleatum* exhibited an even greater tumor burden compared to mice not inoculated with *F. nucleatum*, indicating a tumor-promoting effect. This finding conflicts with the interpretation presented in both the results and discussion sections.

(3) Although the authors acknowledge that *F. nucleatum* may have tumor context-specific roles in regulating NK cell responses, it is unclear why they chose a breast cancer model in which *F. nucleatum* has been reported to promote tumor growth. A more appropriate choice would have been the well-established preclinical oral cancer model, such as the 4-nitroquinoline 1-oxide (4NQO)-induced oral cancer model in C57BL/6 mice, which would more directly relate to HNSC biology.

(4) Since RadD of F. nucleatum can bind to both Siglec-7 and NKp46 on NK cells, exerting opposing functional effects, the expression profiles of both receptors on intratumoral NK cells should be evaluated. This would clarify the balance between activating and inhibitory signals in the tumor microenvironment and provide a more mechanistic explanation for the observed tumor context-dependent outcomes.

---

## [Author Response]

**Reviewer #1 (Public review):**
Major Concerns:(1) Lack of Direct Evidence for RadD-NKp46 InteractionThe central claim that RadD interacts with NKp46 is not formally demonstrated. A direct binding assay (e.g., Biacore, ELISA, or pull-down with purified proteins) is essential to support this assertion. The absence of this fundamental experiment weakens the mechanistic conclusions of the study.

The reviewer is correct. Direct assays are currently quite impossible because RadD is huge protein and it will take years to purify it. Instead, we used immunoprecipitation assays using NKp46-Ig (Author response images 1 and 2). Fusobacteria were lysed using RIPA buffer, and the lysates were centrifuged twice to separate the supernatant from the pellet (which contains the bacterial membranes). The resulting lysates were incubated overnight with 2.5 µg of purified NKp46 and protein G-beads. After thorough washing, the bound proteins were placed in sample buffer and heated at 95 °C for 8 minutes. The eluates were run on a 10% acrylamide gel and visualized by Coomassie blue staining. As can be seen the NKp46-Ig was able to precipitate protein band around 350Kd in both F. *polymorphum ATCC10953* (Author response image 1) and in *F. nucleatum ATCC23726* (Author response image 2).

**Author response image 2. sa3fig2:** 

(2) Figure 2: Binding Specificity and Bacterial StrainsA CEACAM1-Ig control should be included in all binding experiments to distinguish between specific and non-specific Ig interactions. There is differential Ig binding between strains ATCC 23726 and 10953. The authors should quantify RadD expression in each strain to determine if the difference in binding is due to variation in RadD levels.

No significant difference in mCEACAM-1-Ig binding was observed across multiple independent experiments. Author response image 3 shows a representative histogram showing mCEACAM-1-Ig binding to *F. nucleatum* ATCC 23726 and *F. polymorphum* ATCC 10953. Comparable binding levels were detected in both bacterial species (upper histogram). Similarly, NKp46-Ig and Ncr1-Ig fusion proteins exhibited comparable binding patterns (lower histogram). It is currently not possible to quantify RadD expression directly, as no anti-RadD antibody is available.

**Author response image 3. sa3fig3:** 

(3) Figure 3: Flow Cytometry Inconsistencies and Missing ControlsWhat do the FITC-negative, Ig-negative events represent? The authors should clarify whether these are background signals, bacterial aggregates, or debris.

We now present the gating strategy used in these experiments (Author response image 4). Fusion negative Ig samples were the bacterial samples stained only with the secondary antibody APC (anti-human AF647). The TITC-negative represent unlabeled bacteria.

**Author response image 4. sa3fig4:** 

Panel B, CEACAM1-Ig binding appears markedly increased compared to WT bacteria. The reason for this enhancement should be discussed-does it reflect upregulation of the bacterial ligand or an artifact of overexpression? Fluorescence compensation should be carefully reviewed for the NKp46/NCR1-Ig binding assays to ensure that the signals are not due to spectral overlap or nonspecific binding. Importantly, binding experiments using the FadI/RadD double knockout strain are missing and should be included. This control is essential.

We don’t know why expression of CEACAM1-Ig binding is increased. Indeed, it will be nice to have the FadI/RadD double knockout strain which we currently don’t have.

In Panel E, the basis for calculating fold-change in MFI is unclear. Please indicate the reference condition to which the change is normalized.

The mean fluorescence intensity (MFI) fold change was calculated by dividing the MFI obtained from staining with the fusion proteins by the MFI of the corresponding secondary antibody control (bacteria incubated without fusion proteins).

(4) Figure 4: Binding Inhibition and Receptor SensitivityPanel A lacks representative FACS plots and is currently difficult to interpret.

Fusobacteria binding to CEACAM-1, NKp46, and NCR1 fusion proteins was tested in the presence of 5 and 10 mM L-arginine (Author response image 5). L-arginine inhibited the binding of NKp46-Ig and NCR1-Ig, whereas no effect was observed on CEACAM-1-Ig binding.

**Author response image 5. sa3fig5:** 

Differences in the sensitivity of human vs. mouse NKp46 to arginine inhibition should be discussed, given species differences in receptor-ligand interactions.

Ncr1, the murine orthologue of human NKp46, shares approximately 58% sequence identity with its human counterpart (1). The observed differences in arginine-mediated inhibition of bacterial binding between mouse and human NKp46 might stem from structural differences or distinct posttranslational modifications, such as glycosylation. Indeed, prediction algorithms combined with high-performance liquid chromatography analysis revealed that Ncr1 possesses two putative novel O-glycosylation sites, of which only one is conserved in humans (2).

References

(1) Biassoni R., Pessino A., Bottino C., Pende D., Moretta L., Moretta A. The murine homologue of the human NKp46, a triggering receptor involved in the induction of natural cytotoxicity. Eur J Immunol. 1999 Mar; 29(3).

(2) Glasner A., Roth Z., Varvak A., Miletic A., Isaacson B., Bar-On Y., Jonjić S., Khalaila I., Mandelboim O. Identification of putative novel O-glycosylations in the NK killer receptor Ncr1 essential for its activity. Cell Discov. 2015 Dec 22; 1:15036.

What are the inhibition results using *F. nucleatum* strains deficient in FadI?

The inhibition pattern observed in the *F. nucleatum* ΔFadI mutant was comparable to that of the wild-type strain (Author response image 6). When cultured under identical conditions and exposed to increasing concentrations of arginine (0, 5, and 10 mM), the *F. nucleatum* ΔFadI strain also demonstrated a dose-dependent reduction in binding to NKp46 and Ncr1.

**Author response image 6. sa3fig6:** 

In Panel B, CEACAM1-Ig and RadD-deficient bacteria must be included as negative controls for binding specificity upon anti-NKp46 blocking.

We appreciate the request to include CEACAM1-Ig and RadD-deficient bacteria as negative controls for specificity under anti-NKp46 blocking. We don’t not think it is necessary since the 02 antibody is specific for NKp46, we used other anti0NKp46 antibodies that did not block the interaction and an irrelevant antibofy, we showed that arginine produced a dose-dependent reduction in NKp46/Ncr1 binding, consistent with an arginine-inhibitable RadD interaction already shown in our manuscript (Fig. 4A). The ΔRadD strains we used already demonstrate loss of NKp46/Ncr1 binding and loss of NK-boosting activity (Figs. 3, 5). Collectively, these data establish that NKp46/Ncr1 recognition of a high-molecular-weight ligand consistent with RadD is specific and functionally relevant.

Figure 5: Functional NK Activation and Tumor KillingIn Panels B and C, the key control condition (NK cells + anti-NKp46, without bacteria) is missing. This is needed to evaluate if NKp46 recognition is involved in tumor killing. The authors should explicitly test whether pre-incubation of NK cells with bacteria enhances their anti-tumor activity.

No significant difference in NK cell cytotoxicity was observed between untreated NK cells and NK cells incubated with anti-NKp46 antibody in the absence of bacteria. Therefore, the NK + anti-NKp46 (O2) group was included as an additional control alongside the other experimental conditions shown in Figures 5b and 5c, and is presented in Author response image 7 below.

**Author response image 7. sa3fig7:** 

Could bacteria induce stress signals in tumor cells that sensitize them to NK killing? This distinction is critical.

It remains unclear whether the bacteria induce stress-related signals in tumor cells that render them more susceptible to NK cell–mediated cytotoxicity.

(6) Figure 5D: Mechanism of Peripheral ActivationIt is suggested that contact between bacteria and NK cells in the periphery leads to their activation. Can the authors confirm whether this pre-activation leads to enhanced killing of tumor targets, or if bacteria-tumor co-localization is required? The literature indicates that *F. nucleatum* localizes intracellularly within tumor cells. If so, how is RadD accessible to NKp46 on infiltrating NK cells?

We do not expect that pre-activation of NK cells with bacteria would enhance their tumor-killing capacity. In fact, when NK cells were co-incubated with bacteria, we occasionally observed NK cell death. Although *F. nucleatum* can reside intracellularly, bacterial entry requires prior adhesion to tumor cells. At this stage—before internalization—the bacteria are accessible for recognition and binding by NK cells.

(8) Figure 5E and In Vivo RelevanceSurprisingly, *F. nucleatum* infection is associated with increased tumor burden. Does this reflect an immunosuppressive effect? Are NK cells inhibited or exhausted in infected mice (TGIT, SIGLEC7...)? If NK cell activation leads to reduced tumor control in the infected context, the role of RadD-induced activation needs further explanation. RadD-deficient bacteria, which do not activate NK cells, result in even poorer tumor control. This paradox needs to be addressed: how can NK activation impair tumor control while its absence also reduces tumor control?

Siglec-7 lacks a direct orthologue in mice, and neither mouse TIGIT nor CEACAM1 bind *F. nucleatum*. The increased tumor burden observed in infected mice may therefore result from bacterial interference with immune cell infiltration and accumulation within the tumor microenvironment (Parhi, L., Alon-Maimon, T., Sol, A. et al. Breast cancer colonization by *Fusobacterium nucleatum* accelerates tumor growth and metastatic progression. Nat Commun 11, 3259 (2020)). Consequently, the NK cells that do reach the tumor site can recognize and kill *F. nucleatum*–bearing tumor cells through RadD–NKp46 interactions. In the absence of RadD, this recognition is impaired, leading to reduced NK-mediated cytotoxicity and increased tumor growth.

(9) NKp46-Deficient Mice: InconsistenciesIn Ncr1⁻/⁻ mice, infection with WT or RadD-deficient *F. nucleatum* has no impact on tumor burden. This suggests that NKp46 is dispensable in this context and casts doubt on the physiological relevance of the proposed mechanism. This contradiction should be discussed more thoroughly.

Ncr1 is also directly involved in mediating NK cell–dependent killing of tumor cells, even in the absence of bacterial infection. Therefore, in Ncr1-deficient mice, *F. nucleatum* has no additional effect on tumor progression (Glasner, A., Ghadially, H., Gur, C., Stanietsky, N., Tsukerman, P., Enk, J., Mandelboim, O. Recognition and prevention of tumor metastasis by the NK receptor NKp46/NCR1. J Immunol. 2012).

**Reviewer #2 (Public review):**
Weaknesses:(1) A previous study by this group (PMID: 38952680) demonstrated that RadD of *F. nucleatum* binds to NK cells via Siglec-7, thereby diminishing their cytotoxic potential. They further proposed that the RadD-Siglec-7 interaction could act as an immune evasion mechanism exploited by tumor cells. In contrast, the present study reports that RadD of *F. nucleatum* can also bind to the activating receptor NKp46 on NK cells, thereby enhancing their cytotoxic function.

Siglec-7 lacks a direct orthologue in mice, and neither mouse TIGIT nor CEACAM1 bind *F. nucleatum*. In contrast, NKp46 and its murine homologue, Ncr1, both recognize and bind the bacterium.

While *F. nucleatum*-mediated tumor progression has been documented in breast and colon cancers, the current study proposes an NK-activating role for *F. nucleatum* in HNSC. However, it remains unclear whether tumor-infiltrating NK cells in HNSC exhibit differential expression of NKp46 compared to Siglec-7. Furthermore, heterogeneity within the NK cell compartment, particularly in the relative abundance of NKp46⁺ versus Siglec-7⁺ subsets, may differ substantially among breast, colon, and HNSC tumors. Such differences could have been readily investigated using publicly available single-cell datasets. A deeper understanding of this subset heterogeneity in NK cells would better explain why *F. nucleatum* is passively associated with a favorable prognosis in HNSC but correlates with poor outcomes in breast and colon cancers.

Currently, there are no publicly available single-cell datasets suitable for characterizing NK cell heterogeneity in the context of *F. nucleatum* infection—particularly regarding the expression of Siglec-7, NKp46, or CEACAM1 and their potential association with poor clinical outcomes in breast, head and neck squamous cell carcinoma (HNSC), or colorectal cancer (CRC). Furthermore, no RNA-seq datasets are available for breast cancer cases specifically associated with *F. nucleatum* infection and poor prognosis. Therefore, we analyzed bulk RNA expression datasets for Siglec-7 and CEACAM1 and evaluated their associations with HNSC and CRC using the same patient databases utilized in our manuscript (Author response image 8). No significant differences in Siglec-7 expression were detected between HNSC and CRC samples (Author response image 8A). Although CEACAM1 mRNA levels did not differ between *F. nucleatum*–positive and –negative cases within either cancer type, its overall expression was higher in CRC compared to HNSC (Author response image 8B).

**Author response image 8. sa3fig8:** 

(2) The in vivo tumor data (Figure 5D-F) appear to contradict the authors' claims. Specifically, Figure 5E suggests that WT mice engrafted with AT3 breast tumors and inoculated with WT *F. nucleatum* exhibited an even greater tumor burden compared to mice not inoculated with *F. nucleatum*, indicating a tumor-promoting effect. This finding conflicts with the interpretation presented in both the results and discussion sections.

Siglec-7 lacks a direct orthologue in mice, and neither mouse TIGIT nor CEACAM1 bind *F. nucleatum*. The increased tumor burden observed in infected mice may therefore result from bacterial interference with immune cell infiltration and accumulation within the tumor microenvironment (Parhi, L., Alon-Maimon, T., Sol, A. et al. Breast cancer colonization by *Fusobacterium nucleatum* accelerates tumor growth and metastatic progression. Nat Commun 11, 3259 (2020)). Consequently, the NK cells that do reach the tumor site can recognize and kill *F. nucleatum*–bearing tumor cells through RadD–NKp46 interactions. In the absence of RadD, this recognition is impaired, leading to reduced NK-mediated cytotoxicity and increased tumor growth.

(3) Although the authors acknowledge that *F. nucleatum* may have tumor context-specific roles in regulating NK cell responses, it is unclear why they chose a breast cancer model in which *F. nucleatum* has been reported to promote tumor growth. A more appropriate choice would have been the well-established preclinical oral cancer model, such as the 4-nitroquinoline 1-oxide (4NQO)-induced oral cancer model in C57BL/6 mice, which would more directly relate to HNSC biology.

The tumor model we employed is, to date, the only model in which *F. nucleatum* has been shown to exert a measurable effect, which is why we selected it for our study (Parhi, L., Alon-Maimon, T., Sol, A. et al. Breast cancer colonization by *Fusobacterium nucleatum* accelerates tumor growth and metastatic progression. Nat Commun. 2020; 11: 3259). We have not tested the 4-nitroquinoline-1-oxide (4NQO)–induced oral cancer model, and we are uncertain whether its use would be ethically justified.

(4) Since RadD of *F. nucleatum* can bind to both Siglec-7 and NKp46 on NK cells, exerting opposing functional effects, the expression profiles of both receptors on intratumoral NK cells should be evaluated. This would clarify the balance between activating and inhibitory signals in the tumor microenvironment and provide a more mechanistic explanation for the observed tumor context-dependent outcomes.

This question was answered in Author response image 8 above.